# 3D-Printed Poly (P-Dioxanone) Stent for Endovascular Application: In Vitro Evaluations

**DOI:** 10.3390/polym14091755

**Published:** 2022-04-26

**Authors:** Junlin Lu, Xulin Hu, Tianyu Yuan, Jianfei Cao, Yuanli Zhao, Chengdong Xiong, Kainan Li, Xun Ye, Tao Xu, Jizong Zhao

**Affiliations:** 1Beijing Tiantan Hospital, Department of Neurosurgery, Capital Medical University, Beijing 100070, China; ljl147258@outlook.com (J.L.); zhaoyuanli@bjtth.org (Y.Z.); 2Clinical Medical College & Affiliated Hospital of Chengdu University, Chengdu University, Chengdu 610081, China; huxulin1993@163.com (X.H.); likainan1961@126.com (K.L.); 3State Key Laboratory for Turbulence and Complex Systems, Department of Mechanics and Engineering Science, College of Engineering, Peking University, Beijing 100871, China; tyyuan@pku.edu.cn; 4School of Materials and Environmental Engineering, Chengdu Technology University, Chengdu 610041, China; cjf@cdtu.edu.cn; 5Beijing Translational Engineering Enter for 3D Printer in Clinical Neuroscience, Beijing 100070, China; 6Chengdu Institute of Organic Chemistry, Chinese Academy of Sciences, Chengdu 610041, China; xiongcdcioc@163.com; 7Department of Mechanical Engineering, Tsinghua University, Beijing 100084, China; 8Bio-Intelligent Manufacturing and Living Matter Bioprinting Center, Research Institute of Tsinghua University in Shenzhen, Tsinghua University, Shenzhen 518057, China; 9East China Institute of Digital Medical Engineering, Shangrao 334000, China

**Keywords:** 3D printing, intracranial aneurysm, poly (p-dioxanone), bioresorbable stents, endothelization

## Abstract

Rapid formation of innovative, inexpensive, personalized, and quickly reproducible artery bioresorbable stents (BRSs) is significantly important for treating dangerous and sometimes deadly cerebrovascular disorders. It is greatly challenging to give BRSs excellent mechanical properties, biocompatibility, and bioabsorbability. The current BRSs, which are mostly fabricated from poly-l-lactide (PLLA), are usually applied to coronary revascularization but may not be suitable for cerebrovascular revascularization. Here, novel 3D-printed BRSs for cerebrovascular disease enabling anti-stenosis and gradually disappearing after vessel endothelialization are designed and fabricated by combining biocompatible poly (p-dioxanone) (PPDO) and 3D printing technology for the first time. We can control the strut thickness and vessel coverage of BRSs by adjusting the printing parameters to make the size of BRSs suitable for small-diameter vascular use. We added bis-(2,6-diisopropylphenyl) carbodiimide (commercial name: stabaxol^®^-1) to PPDO to improve its hydrolytic stability without affecting its mechanical properties and biocompatibility. In vitro cell experiments confirmed that endothelial cells can be conveniently seeded and attached to the BRSs and subsequently demonstrated good proliferation ability. Owing to the excellent mechanical properties of the monofilaments fabricated by the PPDO, the 3D-printed BRSs with PPDO monofilaments support desirable flexibility, therefore offering a novel BRS application in the vascular disorders field.

## 1. Introduction

Intracranial aneurysms are pathological dilations at cerebral arteries that affect 3~5% of the adult population and cause substantial morbidity and mortality rates [1]. Microsurgical aneurysm clipping and endovascular stent-assisted coiling are two primary treatment strategies due to their effectiveness in blocking the flow of the aneurysm and mitigating the risk for future aneurysm rupture, and they offer the possibility of subarachnoid hemorrhage prevention [2]. Numerous studies have demonstrated that short-term and long-term outcomes are significantly better with endovascular stent-assisted coiling than microsurgical clipping. Hence, endovascular therapy has gradually become the first choice for intracranial aneurysms, especially since the introduction of the flow diversion device into clinical practice [3]. However, compared to microsurgical clipping, the endurance of endovascular therapy remains uncertain because of aneurysm recurrence, as well as local thrombus formation and restenosis [4]. Inevitable vessel wall injury after stenting activates the inflammatory and vascular repair responses, which induce the adhesion of blood platelets as well as the migration and proliferation of smooth muscle cells, leading to delayed reendothelialization, in-stent restenosis, and late stent thrombosis [5,6].

In addition, as an operative implant device in surgery, the traditional long-term metallic stent has some important long-term disadvantages, including inflammation from long-term metallic implants on account of the persistent chronic foreign body reaction, as well as limiting normal vasomotion and adaptive arterial remodeling. In addition, thanks to the inevitable vessel wall injury and metallic implants, patients have to undergo antiplatelet therapy after the procedure and have a higher risk of bleeding triggered by the long-term intake of antiplatelet drugs. After vascular injury, the early establishment of a functional endothelial layer occurs between three and six months. This has been proven to contribute to the prevention of thrombus formation and neointimal proliferation [7,8], as it was suggested that patients take dual antiplatelet treatment for three to six months after placement of the stents.

As alternative implantable stents, bioresorbable stents (BRSs) are promising tools in the field of endovascular intervention; they are bioresorbable and biocompatible, providing supporting and anti-stenosis effects in the short-term and then gradually disappearing after vessel endothelialization to leave behind only the native vessel. At present, poly-l-lactide (PLLA) is the most widely used material of polymeric BRS [9]. However, the stiffness and the viscoelastic characterization of polymers contribute to the insufficient mechanical efficacy under consistent external loads [10,11,12], which might result in acute stent recoil, leading to in-stent restenosis [13]. These conventional PLLA BRS devices rely on significant plastic deformation to resist recoiling, introducing crimping and expanding loads and residual stress in the stents when implanted [14]. In fact, anisotropic loads and stress may increase the ultimate tensile strength of the material, resulting in strut fracture [15]. Moreover, the in vivo acid degradation products of PLLA also trigger inflammation and rejection in the body, which restrict its application in the biomedical field [16,17]. Coronary artery disease usually presents only ischemic lesions, while cerebrovascular disease includes both ischemic and hemorrhagic lesions. The time to complete resorption of most PLLA BRSs for coronary revascularization is more than 2 years, which may be too long for cerebrovascular revascularization with hemorrhagic lesions. In addition, cerebrovascular BRSs is more tortuous than cardiovascular BRSs. The poor compliance of the current PLLA BRSs may make surgical procedures more difficult. Thus, we aim to overcome the current limitations among BRSs.

Poly(p-dioxanone) (PPDO), which was certified by the FDA as safe for humans, possesses the unique properties of excellent biodegradability, biocompatibility, and bioabsorbability to achieve clinical requirements. Compared to the low polymer modulus and mechanical strength of PLLA, as a semicrystalline polymer, PPDO has good mechanical properties with a strength at break close to 46 MPa and an ultimate elongation ranging from 500% to 600% [18,19,20,21]. Such mechanical properties can provide effective radial support and longitudinal tensile tension for the stents. Moreover, it is generally assumed that a functional endothelial layer is established after the implantation of vascular stents between three and six months [22,23,24]. PPDO presents a matched degradation time of approximately 90 days [25,26,27]. As the endothelial cells proliferate and differentiate, the PPDO stents degrade and are absorbed gradually in the body, effectively promoting blood vessel regeneration and reducing postoperative complications, such as inflammation, in-stent restenosis, and late stent thrombosis. Correspondingly, PPDO exhibits a promising alternative for the fabrication of absorbable stents for vascular surgical operation. Previous studies validated the potential of PPDO monofilaments for the manufacture of knitted stents [28,29]. The mechanical properties of the knitted stents differed from those of the filaments treated under various conditions. Based on the lack of research about PPDO braided devices, the characteristics of PPDO stents must be analyzed to guide their design and manufacture. However, PPDO is prone to degradation in vivo, and directly using PPDO as the material for BRSs may not meet degradation time requirements in clinical applications. According to the existing literature, after only 4 weeks in a phosphate buffer solution of pH 7.4, PPDO turned out to be brittle with only 5% of its initial maximum breaking elongation and lost 73% of its initial tensile strength. Many fillers have been added to PDDO to prepare composites to improve its hydrolytic stability. Bis-(2,6-diisopropylphenyl) carbodiimide (commercial name: stabaxol^®^-1), a monogroup aromatic carbodiimide compound, has received a lot of attention on account of its high stability and good biocompatibility. Zhao et al. found that stabaxol^®^-1 retarded the hydrolysis degradation of PPDO and enhanced its hydrolytic stability in a small amount of filler (0.3~1.2%) [19]. The reason for the enhancement is that stabaxol^®^-1 can react with terminal carboxyl groups to generate stable N-acylurea and reduce the autocatalytic activity of acid species originating from the breaking of ester bonds [19].

Digital subtraction angiography (DSA) and computed tomography angiography (CTA) provide a suitable noninvasive approach to assess vascular anatomy. The operator assesses the suitable stents for treatment according to the length of the lesion provided by imaging. However, not all patients can find stents that match the length of their lesions. Therefore, the combination of DSA or CTA imaging and 3D printing has produced an innovative, inexpensive, personalized, and quickly reproducible artery stent [30]. It can provide personalized stents suitable for the lesion length for patients receiving endovascular therapy.

In this study, to combine the advantages of BRS and 3D printing techniques, a novel type of PPDO 3D-printed vascular stent incorporated with stabaxol^®^-1 was fabricated for the first time. Figure 1 is a conceptual diagram of the individualized customized printing of BRSs. The length and diameter of the target vessel were measured by CT imaging and the obtained parameters were imported into the 3D printer. Using PPDO/stabaxol^®^-1 composite with different proportions of stabaxol^®^-1 as ink, the corresponding BRSs were individually printed. In addition, the mechanical properties, in vitro degradation, and biocompatibility of the bioresorbable stent were investigated to examine the potential of this stent for biomedical vascular repair applications.

## 2. Materials and Methods

### 2.1. Sample Preparation

PPDO (Evonik China Co., Ltd., Shanghai, China) was completely dissolved in dichloromethane (*w*/*v* = 1:20, 40 mL, 300 rpm) at room temperature, then additional stabaxol^®^-1 powder (*w*/*v* = 0, 0.3%, 0.6 %) was added. These proportions were derived from the existing literature, which revealed that 0.6% stabaxol^®^-1 retarded hydrolysis of PPDO significantly but when content was higher than 0.6%, the introduction of stabaxol^®^-1 gave rise to the complete end-capping of the active chain end of the polymer and deteriorated the mechanical properties of composites [31]. The mixture was then subjected to sonication for 30 min until it was uniformly dispersed. Then, the mixture was slowly stirred at 300 rpm to facilitate the volatilization of dichloromethane. After that, the PPDO containing stabaxol^®^-1 (abbreviated to PPDO/stabaxol^®^-1) was extensively washed and dried in a vacuum at room temperature. After washing and drying, the status of the PPDO/stabaxol^®^-1 was a chunk. We then ground the chunk to create PPDO/stabaxol^®^-1 pellets.

### 2.2. Fabrication of PPDO and PPDO/Stabaxol^®^-1 Stents

For this study, the composite stents were prepared using a 3D printer (Livprint Norm, Medprin, Guangzhou, China). The printing instrumentation consisted of a 3D bioprinting system equipped with a three-axis X-Y-Z translation stage, dispenser, nozzle, compression/heat controller, and software system. PPDO and PPDO/stabaxol^®^-1 pellets were melted at 180 °C in a heated dispenser. The nozzle diameter was 160 μm. After the PPDO and PPDO/stabaxol^®^-1 pellets were melted, a continuous air pressure of 180 kPa was applied to the dispenser, and strands of molten PPDO were rolled onto a 3 mm diameter rod. After the printing, we put the rolling rod in the refrigerator to make the molten strands cool and form. According to the experience of our laboratory technician, we could easily remove the flexible stent from the rolling rod by spraying it with ethanol. Finally, the 3D-printed PPDO stents were dried at 37 °C for 2 days to remove moisture from the above process. We devised a formula to ensure the controllability of the 3D-printed stents and the consistency of each 3D-printed stent, which states that for a given radius of the rod (r), strut angle (θ) is proportional to the horizontal movement speed of the nozzle (v) and mandrel speed of the rod (ω):(1)$$\tan\frac{\pi -\theta }{2}=\frac{\omega r}{v}.$$

### 2.3. Morphological and Structural Characterization

The structure of the PPDO BRS, as well as the morphologies of the PPDO/stabaxol^®^-1 BRS, were characterized. Specimens were freeze-dried and coated with a thin gold layer by sputtering, then visualized under a field emission scanning electron microscope (SEM, Phenom Pro, Phenom Scientific China Co., Ltd., Shanghai, China), with an accelerating voltage of 10.0 kV. Chemical information for the indicated samples, treated after the KBr pressed pellet method, was obtained through FTIR spectroscopy (Thermo Fisher, Nicolet 6700, Waltham, MA, USA) with wavelength ranging from 3500 to 500 cm^−1^. Crystallinity and glass transition temperatures (T*g*) of PPDO and composite PPDO/stabaxol^®^-1 described in this study were determined by differential scanning calorimetry (DSC, Q20, TA Instrument, New Castle, DE, USA). Samples (5–10 mg) were analyzed at a heating rate of 10 °C/min in the temperature range −80 to 200 °C using a high-purity nitrogen purge.

### 2.4. Mechanical Property Test

Radial compression tests were conducted for evaluating the mechanical properties of the PPDO and the PPDO/stabaxol^®^-1 BRSs, using a universal testing machine (KT23.104, Guangzhou Kiatest equipment, Co. Ltd., Guangzhou, China) under the international test standard (ISO25539-2-2012). The radial compression was defined as an effective method for evaluating stent characteristics in response to localized compressive load, including the ability of stents to resist the external force and recover their original geometry (I. Standards, n.d.). Briefly, the presser foot first went down to compress stents with the speed of 1 mm/min until 50% deformation of outer diameter. Forces were recorded by a sensor in the presser foot, connected to the computer. The presser foot went up to unload stents after keeping compressed for 30 s. During unloading for 30 s, we observed whether the stents could return to the initial diameter. Load-displacement curves were obtained, and the functional properties of stents were calculated accordingly. All tests were conducted under standard environmental conditions (20 ± 1 °C, RH 65 ± 2%).

The resulting compression modulus of PPDO BRSs (*n* = 5) was calculated by the following formula:Compression modulus (MPa) = Stress difference (σ2 − σ1)/Strain difference (ε2 − ε1 = 0.0025 − 0.0005)
where σ1 is the corresponding stress value measured when ε1 is set as 0.0005 and σ2 represents the stress value when ε2 is 0.0025.

### 2.5. Finite Element Analysis

As a comparison with the compression test, we conducted a numerical simulation of the mechanical behavior of the PPDO stents through the finite element method. The software Abaqus Standard Solver (version 2021) was chosen as the platform.

The configurations of the PPDO stents and two parallel compression plates were directly created via Abaqus tools. Since the compression plates had a much larger elastic modulus than the PPDO stent, the plates were modeled by the analytical rigid body element to improve the computational efficiency. For the analytical rigid body element, the reference point was positioned at the centroid of each plate, while noting that the material property and the mesh of it need not be specified.

According to previous research [32], the PPDO monofilaments that constitute the stents are widely recognized as hyperelastic incompressible materials. There are plenty of models in regard to the hyperelasticity.

By neglecting both the degradation and the temperature dependence of the elastic modulus of the PPDO monofilament, we can directly use the simple neo-Hookean strain energy potential Ws to estimate the strain energy during the large deformation of PPDO monofilaments, i.e.,
(2)Ws=C10(I1−3)
where C10 is a material parameter, and I1 is the first principal invariant of the right Cauchy–Green deformation tensor C∈ℝsym3×3 or the left Cauchy–Green deformation tensor B∈ℝsym3×3. 

In other words,
(3)$$I_{1}=\mathrm{Trace}\left(C\right)=\mathrm{Trace}\left(B\right)$$

C and B can be expressed in terms of the deformation gradient F∈ℝ3×3, i.e.,
(4)C=FTF and B=FFT
with F defined by
(5)dx=FdX
where dx is the infinitesimal line element for the current configuration and dX is that for the reference configuration.

By polar decomposition, F can be expressed as
(6)F=RU=VR
where R∈ℝ3×3 is an orthogonal tensor that physically represents the rotation of dX**,** and U∈ℝ3×3 and V∈ℝ3×3 are the right and left stretch tensors, respectively. The eigenvalues of U and V are the same, i.e., λ1,λ3,λ3. These eigenvalues are stretch ratios along the Lagrange principal directions (for U) or Euler principal directions (for V). In conclusion,
(7)I1=λ12+λ22+λ32

Therefore
(8)$$W^{s}=C_{10}\left(\lambda_{1}{}^{2}+\lambda_{2}{}^{2}+\lambda_{3}{}^{2}-3\right)$$

However, to the authors’ best knowledge, there does not exist any specific result on the magnitude of $C_{10}$ for PPDO, which needs to be set initially before the numerical calculation. Hence, we treat it as a fitting parameter associated with the experiments. Since, in the context of linear elasticity and isotropy, $C_{10}=\frac{G}{2},$ where G is shear modulus, then based on one scientific article [33], the magnitude of $C_{10}$ can be preliminarily assumed to be 10 MPa and then adjusted according to the experimental data. 

Considering the incompressibility of PPDO, we used a C3D8H (hybrid formulation) element to mesh the specimen. The size was set as 0.01 mm. Similar to the previous research [29], we used penalty friction formulation with coefficient 0.25 to define the interaction among all surfaces. The PPDO specimen was constrained along its axial direction (y-direction in Figure 3B). There is a relative translational motion of two plates along the compression direction (z-direction in Figure 3B) until the deformation reaches 30%, in which situation the theoretical model can be valid. Other translational or rotational motion of the plate is forbidden.

### 2.6. Hydrolytic Assays

To perform in vitro degradation assays, composite stents (30 mm length × 3 mm diameter) were immersed in phosphate buffer solution (PBS) (10 mL, pH 7.4). Samples were then kept in a 37 °C incubator and weighed at 0, 2, 4, and 8 weeks separately, after washing with deionized water 5 times. The water absorption and mass loss were calculated from the following formulae, respectively:Water absorption (%) = (m_wet_ − m_0_)/m_0_ × 100(9)
Mass loss (%) = (m_0_ − m_1_)/m_0_ × 100(10)
where m_wet_ represents the wet weight of the BRSs investigated in this study, while m_0_ and m_1_ stand for the dry weights of scaffolds before and after the degradation assay.

### 2.7. Cell Culture

Primary human umbilical vein endothelial cells (HUVECs) were used to study the biotoxicity and biocompatibility of PPDO and PPDO/stabaxol^®^-1 BRSs. The primary HUVECs were obtained from ScienCell Research Laboratories, Inc, and cultured in cell culture flasks in a 37 °C, 5% CO_2_ humidified atmosphere incubator. Endothelial culture medium (ScienCell, Carlsbad, CA, USA) complemented with 1% (*v/v*) penicillin/streptomycin (ScienCell, Carlsbad, CA, USA) and FBS (ScienCell, Carlsbad, CA, USA) was used for cell culture and was refreshed every two days. HUVECs were trypsinized with 0.05% T/E (Cat. #0183) at passage 2 and passage 3, and were seeded onto the sterilized PPDO and PPDO/stabaxol^®^-1 BRSs at a cell density of about 1 × 10^5^ cells/mL; we used at least 3 stents in each cell assay.

### 2.8. Cell Viability and Cell Adhesion Assay

After HUVECs had been cultured onto the sterilized PPDO and PPDO/stabaxol^®^-1 BRSs for 1, 3, and 7 days, the viability of HUVECs on each stent was assessed using a LIVE/DEAD Viability/Cytotoxicity Kit (Beyotime Biotechnology, Shanghai, China) for cell staining. HUVECs on the stents were immersed in a working solution containing 2 × 10^−9^ m Calcein AM and 4 × 10^−9^ m Ethidium homodimer-1 for, respectively, labeling live cells (green color) and dead cells (red color) at room temperature for 10 min; they were then washed with PBS three times and subsequently visualized under a fluorescence microscope (Eclipse Ti2-U, Nikon, Japan). Alamar Blue Kit (YEASEN, Shanghai, China) was used to evaluate cell proliferation according to the manufacturer’s instructions. The measurements were performed on day 1, day 3, and day 7 after the cells were seeded on the stents. The optical density (OD) value of the supernatant was read on a microplate reader (Epoch 2, BIOTEK, Winooski, VT, USA) at wavelengths of 570 and 600 nm. Cell seeding efficiency (E_i_) onto PPDO and PPDO/stabaxol^®^-1 BRSs was determined by the ratio of the initial density of seeded cells (D_0_, 1 × 10^5^ cells/cm^2^) to the density of HUVECs on the BRSs after 1 d cell culture (D_i_, assessed at a square range of 300 × 300 μm) according to
E_i_ = D_i_/D_0_.(11)

### 2.9. Statistical Analysis

The statistical results were expressed as mean ± standard deviation. Statistical analysis was carried out using analysis of variance with a one-way ANOVA or Student’s *t*-test. *p* value < 0.001, ***, *p* value < 0.01, **, *p* value < 0.05, *, n.s.: not significant. All figures involving statistics were generated by GraphPad Prism 9.0 (GraphPad Software, LLC).

## 3. Results

### 3.1. PPDO and PPDO/Stabaxol^®^-1

To characterize the morphology of the printed stents and discuss a series of effects of stabaxol^®^-1 addition to the materials due to different proportions, relevant assays were performed. The thermal properties of PPDO and composite PPDO/stabaxol^®^-1 were analyzed with DSC. PPDO exhibited only one glass transition peak at 49.70 °C, whereas crystallization and melting phenomena were undetected, which were also observed in the composite of PPDO/stabaxol^®^-1 (Figure 2C). For the composite group, PPDO/0.3% stabaxol^®^-1 and PPDO/0.6% stabaxol^®^-1 possessed Tg at 53.63 °C and 53.98 °C, respectively. The DSC data indicated that the addition of stabaxol^®^-1 slightly improved the T*g* of PPDO.

The FT-IR spectra of the PPDO and composite PPDO/stabaxol^®^-1 (0.3% and 0.6%) are presented in Figure 2D. The characteristic peak of the carbodiimide group is at 2162.3 cm^−1^, along with three phenyl group characteristic absorption peaks at 1583.9 cm^−1^ (I), 1516.2 cm^−1^ (II), and 1464.7 cm^−1^ (III). The absorption bands of O-H, C=O, and H-O-C(O) -C groups are located at 3503, 1755.3, and 1187.9 cm^−1^, respectively. The peaks at 1622.3 and 1623.0 cm^−1^ denote the characteristic absorption of the phenyl group introduced by the addition of stabaxol^®^-1. The absorption for the carbodiimide group could possibly be disguised by the wide and sharp peak of C=O stretching vibration. As above, it was clear that the position of the characteristic peaks was maintained with the increase in stabaxol^®^-1 content, demonstrating the weak interaction between PPDO and stabaxol^®^-1.

### 3.2. Surface Morphology and Configuration of PPDO and PPDO/Stabaxol^®^-1 Stents

The surface interlacing morphology was observed in PPDO and PPDO/stabaxol^®^-1 stents using microscopy and SEM photography. The structure and surface imaging of PPDO stents are shown in Figure 2A,B, respectively. The surface of PPDO stents was relatively smooth with small pores, demonstrating suitable structural features for the following endothelial cell attachment and confluent endothelial monolayer formation. Both PPDO and PPDO/stabaxol^®^-1 stents have symmetrical structures and the wires form a uniform diamond shape. The spiral wires are parallel with equal intervals. As the PPDO and PPDO/stabaxol^®^-1 BRSs adopted in this research use braiding manufacturing technology, the strut angle and pitch length are not fixed. In the present study, the strut thickness is related with the speed of the nozzle and rotation speed of the rod. We printed the BRSs with a nozzle speed of 5 mm/s and a rotation speed of 17.5 rad/min, which contribute to the strut thickness of 140 μm with vessel coverage of 22%.

### 3.3. Mechanical Properties of PPDO and PPDO/Stabaxol^®^-1 Stents

To fit the natural bending and the pressure of contraction of vessel walls, PPDO and PPDO/stabaxol^®^-1 stents were fabricated with the braiding structure, which provides the stents with good flexibility. The radial compression behavior of PPDO, PPDO/0.3% stabaxol^®^-1, and PPDO/0.6% stabaxol^®^-1 BRSs is presented in Figure 3A. Interestingly, no significant differences were observed among these groups despite their slight variation. In addition, after 30 s unloading the pressure, all BRSs can return to the initial diameter (Appendix A).

### 3.4. Finite Element Analysis

We ultimately found $C_{10}=600\;\mathrm{MPa}$. The deformed configuration of the PPDO specimen is presented in Figure 3C,D. It is emphasized that the unit for von Mises stress is MPa in the above figures. The maximum of the von Mises stress is around 1000 MPa, which reaches the neighborhood of the contact position of internal and external PPDO monofilaments. Furthermore, the simulation results along with the experimental results were plotted (Figure 3E), showing good agreement with each other.

### 3.5. Degradation of PPDO and PPDO/Stabaxol^®^-1 Stent

In this study, hydrolysis of the stents was investigated for up to 8 weeks; pH and mass loss were measured at each time point (0, 2, 4, and 8 weeks). The stable microstructure is the key point for illustrating the effectiveness of BRSs. Surface micrographs of PPDO, PPDO/0.3% stabaxol^®^-1, and PPDO/0.6%stabaxol^®^-1 before and after hydrolysis with different degradation times are shown in Figure 4. PPDO, PPDO/0.3% stabaxol^®^-1, and PPDO/0.6% stabaxol^®^-1 all presented a relatively smooth and clear surface before hydrolysis and after 2 weeks of degradation. Under 37 °C after 4 weeks, some small cracks were exhibited on the surface of PPDO, while few cracks were observed on the surface of PPDO/0.3% stabaxol^®^-1 and PPDO/0.6% stabaxol^®^-1. With further increasing the degradation time to 8 weeks, there were some cracks on the surface of PPDO, PPDO/0.3% stabaxol^®^-1, and PPDO/0.6% stabaxol^®^-1. However, PPDO/0.3% stabaxol^®^-1 and PPDO/0.6% stabaxol^®^-1 were not as fragile as PPDO. The changes in surface morphologies demonstrate that the physical integrity of PPDO/0.3% stabaxol^®^-1 and PPDO/0.6% stabaxol^®^-1 is better than that of PPDO in the degradation process, suggesting that stabaxol^®^-1 can inhibit the hydrolytic degradation of PPDO. Moreover, the water absorption of the BRSs was investigated via weight measurement. Our experiment clearly demonstrated that the addition of stabaxol^®^-1 slightly reduced the rising tendency of the water absorption (Figure 5A). Next, we observed that the addition of stabaxol^®^-1 did extraordinary work, as PPDO/0.3% stabaxol^®^-1 and PPDO/0.6%stabaxol^®^-1 groups showed a significant decrease in mass loss compared to the PPDO group after 8 weeks (Figure 5B). Furthermore, the pH values of PPDO/0.3% stabaxol^®^-1 and PPDO/0.6% stabaxol^®^-1 groups were higher than that of the PPDO group, possibly as a result of fewer acid products being released.

### 3.6. Cytocompatibility and Cell Adhesion of PPDO and PPDO/Stabaxol^®^-1 Stents

To better investigate the cytocompatibility of PPDO and PPDO/stabaxol^®^-1 BRSs, HUVEC attachment on PPDO and PPDO/stabaxol^®^-1 BRSs was statistically analyzed by the determination of cell seeding density (cells/cm^2^) obtained after the cell seeding processes with respect to the density of HUVECs initially seeded onto the stents (1 × 10^5^ cells/cm^2^). Results suggest that HUVECs cultured on each type of stent had similar cell seeding efficiency and cell attachment (Figure 6). Subsequently, HUVECs seeded onto the BRSs were stained by a LIVE/DEAD Viability/Cytotoxicity Kit and visualized under a fluorescence microscope at preset time points. On day 1, HUVECs adhered successfully to the surface and junctions of the bioabsorbable stents and only a few cells were dead, which was not significantly different between PPDO and PPDO/stabaxol^®^-1 BRSs. On days 3 and 7, cells on the bioabsorbable stents increased gradually and spread and covered indicated stents with hardly no new cell death (Figure 7A). Desirable cell viability and cell adhesion status were present for each type of stent at day 7 (Figure 7A). The growth and viability of HUVECs on the stents in the following culture periods were also studied. After 1, 3, and 7 d of culture, the proliferation of HUVECs was statistically analyzed using an Alamar Blue assay. Surprisingly, all stents performed similarly in cell viability, confirming they are all kind to cells and exhibit good biocompatibility (Figure 7B).

## 4. Discussion

With the popularization and application of MRI and CT, the detection rate of intracranial aneurysms is increasing over the years. Endovascular intervention has gradually become the first choice of treatment for intracranial aneurysms. However, the implantation of permanent metal stents may have unpredictable effects on the long-term prognosis of patients. Thus, bioresorbable and biocompatible BRSs sound like a promising tool in the field of endovascular intervention. At present, PLLA is the most widely used material for polymeric BRSs. However, the stiffness and acid degradation products of PLLA restrict its use in clinical applications. Several large randomized controlled trials have demonstrated that the PLLA BRSs lead to worse clinical outcomes compared to metal stents. In addition, the approximately 2 year degradation time of PLLA BRSs may not match the in-stent endothelialization time [34,35]. PPDO has better mechanical properties and a shorter degradation duration compared to PLLA and may be an appropriate material for BRSs [36,37].

In this study, PPDO BRSs were successfully prepared. Moreover, the hydrolytic stability of the BRSs was enhanced by adding stabaxol^®^-1. The compression force of BRSs was slightly increased by adding stabaxol^®^-1, though the difference was not significant enough. Moreover, the addition of stabaxol^®^-1 did not affect the adhesion and proliferation of HUVEC on our BRSs, which contribute to the establishment of the functional endothelial layer. This is important in endovascular devices because the establishment of the functional endothelial layer can reduce the risk of further in-stent restenosis and late stent thrombosis.

It is essential to match the degradation time of BRSs with the time of in-stent endothelialization. Based on previous studies, we added carbodiimide to enhance the hydrolytic stability of the PPDO. The multigroup carbodiimide may give rise to the crosslinking of PPDO, which is not conducive to further processing of PPDO. The stabaxol^®^-1 is a kind of monogroup aromatic carbodiimide compound, with high stability and a suitable melting point, which avoids the disadvantage of multigroup carbodiimide, allowing us to successfully process composite PPDO/stabaxol^®^-1 to fabricate BRSs.

The strut thickness has been reported as an important structure parameter that is significantly associated with stent thrombosis. Although increased strut thickness prevents elastic recoil and increased radial support, it reduces deliverability as it also acutely causes flow disturbances and decreases the neo-intimal area. Strut thickness can strongly affect the speed of endothelial coverage of the struts, which is influenced by shear stress and blood-flow dynamics [38,39]. The uncovered struts have been shown to be one of the major causes of stent thrombosis [9]. Current PLLA BRSs have a strut thickness ranging from 70 μm to 170 μm with a vessel coverage ranging from 20% to 47% [9]. Higher strut thickness is responsible not only for poor deliverability but also higher neo-intimal volumes, possibly leading to flow limitations. It is also associated with poor long-term outcomes: restenosis and stent thrombosis. Thus, the challenge is to have adequate radial support with a low strut thickness. PLLA BRSs having substantially thicker struts than the traditional metal stents, together with malapposition caused by the poor compliance of PLLA, are likely to be the leading cause of stent thrombosis. Conversely, our PPDO BRSs presented thinner struts than PLLA BRSs and macroscopically good flexibility. These advantages might reduce the risk of stent thrombosis and improve the further outcome of patients.

Radial behavior is of great significance in the design of artery stents. The majority of the BRSs studied in the current research field are designed to be applied for cardiovascular interventional therapy, while there are few reports on cerebrovascular interventional therapy. Moreover, PLLA BRSs are mostly laser-cut stents, while the BRSs in this study are similar to braided stents. It has been reported that braided stents present higher coverage and lower radial force compared to laser-cut stents [40,41]. Thus, we evaluated the mechanical properties of PPDO BRSs in this study by judging whether they were within the mechanical range of current commercial products rather than comparing with other BRSs studied in the current research field. Published data demonstrate that the radial force of currently commercialized stents ranges from 2 to 4 N; the mechanical properties of our BRSs fell within the appropriate range for vascular stents [29,42,43]. The radial performance of BRSs is extremely important for evaluating the efficiency of these devices in recovering to their original states after suffering from pulsating pressure and reducing stent recoil. Our data also showed that all BRSs can recover to the original diameter, indicating that our prepared BRSs had good resilience. Given the semi-crystalline structure of the polymer, the monofilaments can be obtained after extrusion and drawn to achieve stents with excellent mechanical properties.

Lin et al. proposed the degradation mechanism of PPDO by observing the change in PPDO sutures in PBS of pH 7.4 at 37 °C [44]. Hydrolysis of PPDO fibers proceeded via two stages: random scission of chain segments located in the amorphous regions of microfibrils and intermicrofibrillar space, followed by stepwise scission of chain segments located in the crystalline regions of microfibrils [44]. The cleavage of the unstable ester bond contributes to the scission of chain segments. The low molecular acid species produced by hydrolysis can further catalyze the scission of ester bonds, accelerating PPDO hydrolysis. Thus, we can enhance the hydrolytic stability of PPDO by suppressing the catalytic activity of acid species. Several efforts have been made to improve the hydrolytic stability of PPDO. Data from Liu et al. indicated that polycarbodiimide can dramatically enhance the hydrolytic stability of PPDO [31]. The enhancement could be attributed to carbodiimide reacting with terminal carboxyl groups of PPDO to generate stable N-acylurea, which reduce the autocatalytic activity of acid species originated from the cleavage of ester bonds. Moreover, it is reported that luminal enlargement occurs between six months and five years after balloon angioplasty for most patients [45]. Meanwhile, the functional endothelial layer is generally established after the implantation of vascular stents between three and six months. Thus, stents in the body existing longer than 6 months may constrain the lumen expansion process. Compared with the degradation time of the PLLA stents, which is longer than 2 years, the degradation time of our PPDO stents is shorter, which is more in line with the normal physiological process.

The speed of endothelial coverage of the struts is influenced by many factors, such as strut thickness, type of polymer, drug type and dose, extent of injury, and type of underlying atherosclerotic disease. PPDO was certified by the FDA with excellent biocompatibility and has been used as a biomaterial for sutures, bone repair devices, and drug delivery systems. However, the formation of a confluent endothelial monolayer on 3D structured BRSs may be harder than on a planar surface, since endothelial cells are unlikely to be homogeneously located onto struts. The results obtained in the present investigation validate the hypothesis that PPDO BRSs could facilitate desirable intracellular connection and the following endothelialization under the 3D cell culture condition, further preventing thrombus formation and neointimal proliferation after implantation.

Although we made some great progress, there are still some limitations in our study. We fabricated novel PPDO BRSs and evaluated their physicochemical properties and biocompatibility in vitro. Further animal experiments to verify the in vivo performance of our novel BRSs is necessary.

## 5. Conclusions

In summary, novel programmed BRSs produced by combining bioabsorbable PPDO polymers and 3D printing have been reported. BRSs consisting of PPDO monofilaments effectively achieve endothelial cell adhesion and degradation after reendothelialization. Through varying the weight of bis-(2,6-diisopropylphenyl) carbodiimide of the BRSs, the degradation stability of PPDO was enhanced accordingly. The BRSs of PPDO exhibited desirable characteristics as 3D cell microenvironments for supporting endothelial cell attachment and growth. By generating firm cell-stent interaction and cell–cell interaction, 3D endothelialization of the stents with braiding PPDO monofilaments and bioabsorbable properties can be effectively promoted. Our methods present a practical approach for personalized customization of small-diameter intravascular stents with the potential to rapidly form a confluent endothelial monolayer in vitro, implying that it is a promising method by which to make small-diameter intravascular stents with improved endothelialization and decreased risk of late stent thrombosis. With the combination of bioabsorbable polymers and 3D printing techniques, this method can be simply expanded to versatile methods for customized manufacturing of intravascular stents of different sizes on demand, thus paving the way to reducing the risk of post-procedure complications of intravascular stenting and improving the outcome of patients.

## Figures and Tables

**Figure 1 polymers-14-01755-f001:**
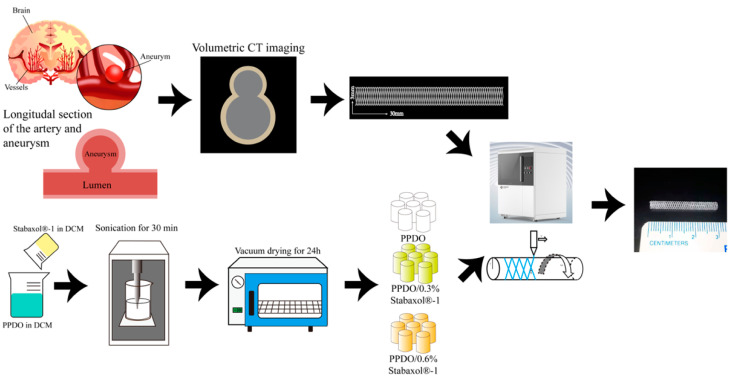
Detailed illustration of preparation process of PPDO stents. The length and diameter of the target vessel were measured by CT imaging and the obtained parameters were imported into the 3D printer. Using PPDO/stabaxol^®^-1 composite with different proportions of stabaxol^®^-1 as ink, the corresponding BRSs were individually printed.

**Figure 2 polymers-14-01755-f002:**
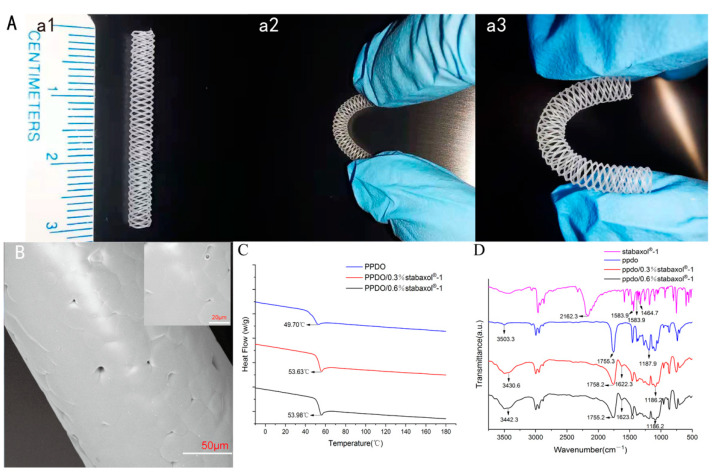
Characterization of PPDO, PPDO/0.3% stabaxol^®^-1, and PPDO/0.6% stabaxol^®^-1 BRSs. (**A**) BRSs with dimensions of 30 mm × 3 mm with significant flexibility. (**B**) Sectional view of PPDO BRS captured by electron microscopy (SEM); ×1100. Scale bar: 50 μm, ×3200. Scale bar: 20 μm. (**C**) DSC and (**D**) FT-IR spectra of indicated BRSs.

**Figure 3 polymers-14-01755-f003:**
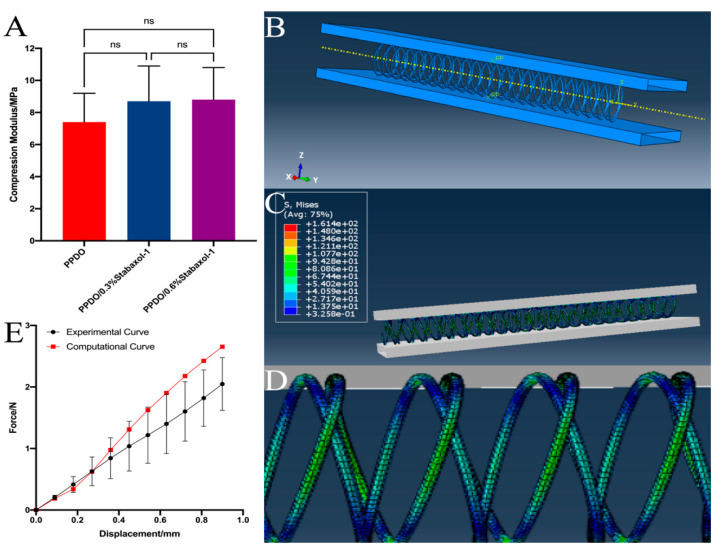
(**A**) Compression forces of PPDO, PPDO/0.3% stabaxol^®^-1, and PPDO/0.6% stabaxol^®^-1 BRSs when compressed to the diameter at 50% of origin. (**B**) The simulation illustrates the finite element analysis results of the PPDO prototype stents. (**C**) The whole and (**D**) local distribution of von Mises stress in the PPDO stents after compression to 30% of their origin diameters. (**E**) Computational and experimental compression force versus displacement curves.

**Figure 4 polymers-14-01755-f004:**
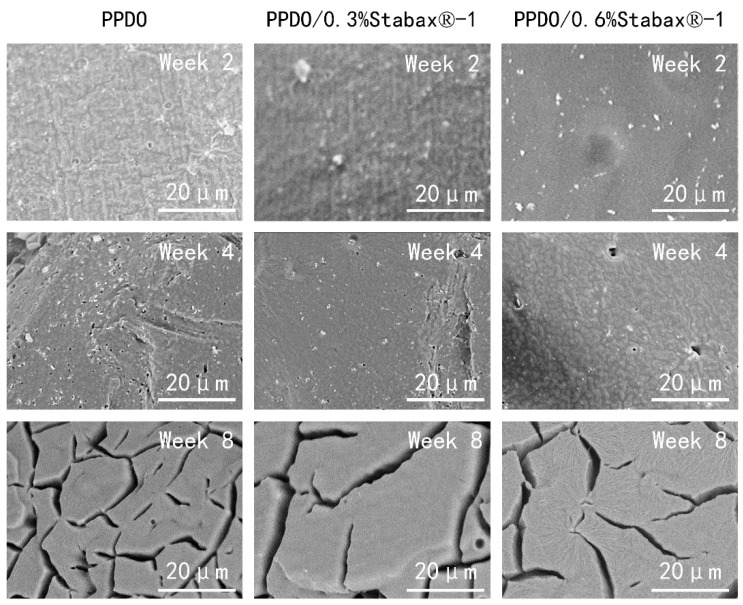
Surface morphology of PPDO, PPDO/0.3% stabaxol^®^-1, and PPDO/0.6% stabaxol^®^-1 from SEM micrographs (×3000) after degradation for 2, 4, and 8 weeks.

**Figure 5 polymers-14-01755-f005:**
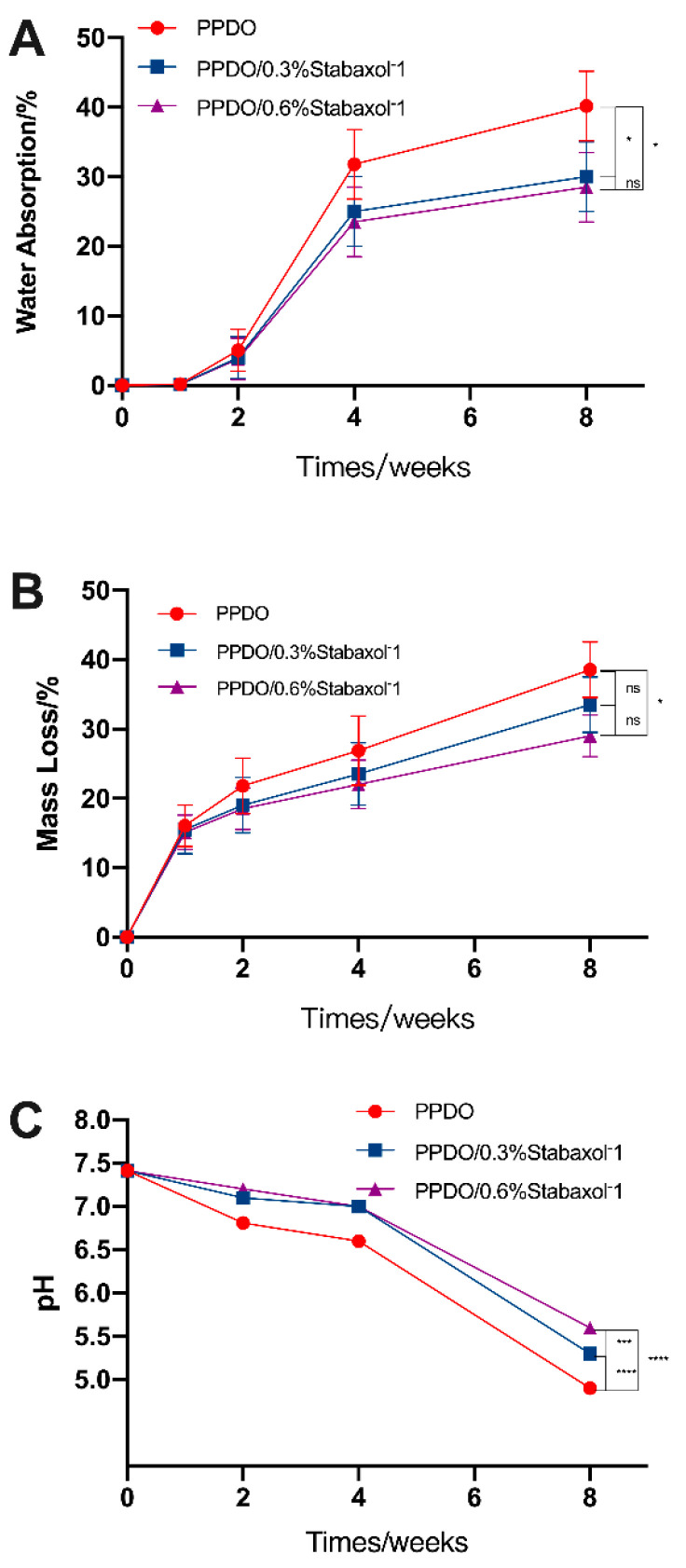
Hydrolysis degradation assays. (**A**) Water absorption was carried out exactly as described in the Materials and Methods section. Compared with PPDO, PPDO/0.3% stabaxol^®^-1 and PPDO/0.6% stabaxol^®^-1 showed lower water absorption after 8 weeks. (**B**) Mass loss was measured and calculated as described in the Materials and Methods section. The mass loss of the PPDO group declined obviously, while that of the PPDO/0.6% stabaxol^®^-1 group decreased less; *p* value < 0.05, *. (**C**) pH monitoring during hydrolysis. pH value was taken and recorded at each time point. The pH value dropped sharply without stabaxol^®^-1; *p* value < 0.001, ***, *p* value < 0.0001, ****.

**Figure 6 polymers-14-01755-f006:**
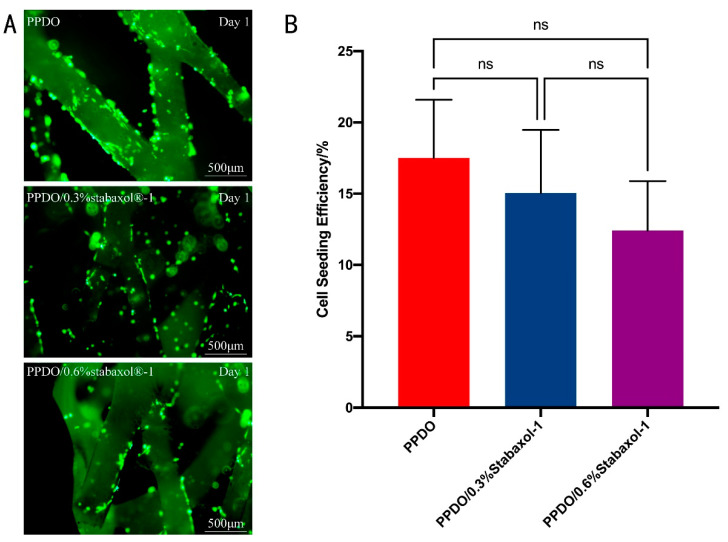
Adhesion of HUVECs to the different BRS groups. (**A**) Inverted fluorescence images of PPDO, PPDO/0.3% stabaxol^®^-1, and PPDO/0.6% stabaxol^®^-1 BRSs on day 1. (**B**) Cell seeding efficiency of HUVECs cultured on the bilayer scaffolds with different weight of stabaxol^®^-1.

**Figure 7 polymers-14-01755-f007:**
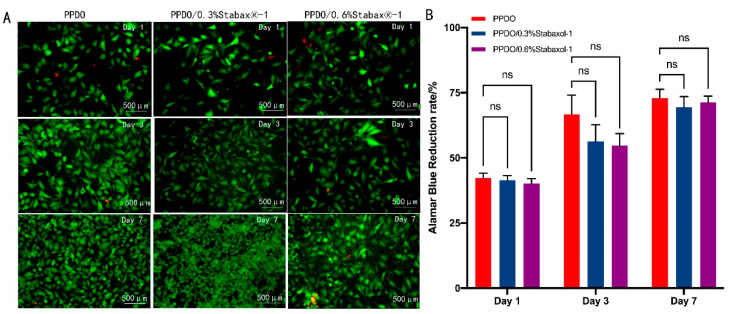
Cytocompatibility of HUVECs on the different BRS groups. (**A**) Merged fluorescent images with live cells stained in green and dead cells stained in red, indicating the viability of HUVECs on PPDO, PPDO/0.3% stabaxol^®^-1, and PPDO/0.6% stabaxol^®^-1 BRSs at various time intervals (1, 3, 7 days). (**B**) Viability of HUVECs cultured on BRSs with different weights of stabaxol^®^-1 after different cell culture time periods.

## Data Availability

The data presented in this study are available on request from the corresponding author.

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
