# Peer review of "3D-Printed Poly (P-Dioxanone) Stent for Endovascular Application: In Vitro Evaluations"

_polymers, 2022, doi:10.3390/polym14091755_

Round 1

Reviewer 1 Report

PPDO/stabaxol®-1BRSs were successfully prepared by 3D printing, and the hydrolytic stability were enhanced compared with PPDO.
1. Discuss more printing procedure with 3D printer and how the authors optimized temperature, pressure, rheological etc.

2. Discuss more practical advantages of PPDO/stabaxol®-1 inks in comparison with other reported approaches.

3. The role of stabaxol®-1 in enhancing the compressive properties and the hydrolytic stability ?

Author Response

Dear Editor and Reviewers from the Editorial Board:

Thank you for suggesting some revisions and thanks for the reviewers’ comments concerning our manuscript entitled “3D-printed poly (p-dioxanone) stent for endovascular application: in vitro evaluations”. (Manuscript ID: polymers-1651615). We are grateful for your positive comments. We have studied the comments carefully and have made revisions accordingly, which we hope will meet with your approval. The revised sections are marked up using the “Track Changes” function in the paper. The point-by-point responses to the reviewer comments are as follows:

Reviewer #1: PPDO/stabaxol®-1BRSs were successfully prepared by 3D printing, and the hydrolytic stability were enhanced compared with PPDO.

Point 1: Discuss more printing procedure with 3D printer and how the authors optimized temperature, pressure, rheological etc.

Response 1: Thanks for your comment. In this study, we fabricated the 3D printed PPDO BRSs successfully. However, the present study focuses on the mechanical properties, hydrolytic stability and biocompatibility of the PPDO BRSs and whether the addition of stabaxol®-1 changes the above properties. As suggested by the reviewer, the printing parameters such as temperature, pressure and rheology during printing are very important in the preparation of scaffolds. We take reviewer’s advice and will further optimize the printing parameters in the follow-up research.

Point 2: Discuss more practical advantages of PPDO/stabaxol®-1 inks in comparison with other reported approaches.

Response 2: Thanks for these comments. We have made revisions to add more discussion and comparison in manuscripts(Section 4, Line 437):“At present, poly-l-lactide (PLLA) is the most widely-used material of polymeric BRS. However, the stiffness and acid degradation products of PLLA restricts its application in the clinical applications. Several large randomized controlled trials demonstrated that the PLLA BRS lead to worse clinical outcomes compared to the metal stents. Also, the nearly 2-year degradation time of PLLA BRS cannot match the in-stent endothelialization time. PPDO has good mechanical properties with a strength at break close to 46 MPa and an ultimate elongation ranging from 500% to 600%, along with a shorter degradation time, which is more suitable for cerebrovascular applications.”

Point 3: The role of stabaxol®-1 in enhancing the compressive properties and the hydrolytic stability ?

Response 3: According to the previous literatures and research, the small amount of stabaxol®-1 filler has been reported to retarded the hydrolysis degradation of PPDO and enhance its hydrolytic stability. The reason for enhancement is that stabaxol®-1 can react with terminal carboxyl groups to generate stable N-acylurea and reduce the autocatalytic activity of acid species originating from the break of ester bonds. We discuss it in the Section 1. Relevant references are as follows:

1.Liu, Z. P., S. D. Ding, Y. J. Sui, and Y. Z. Wang. "Enhanced Hydrolytic Stability of Poly(P-Dioxanone) with Polycarbodiimide." Journal of Applied Polymer Science (2009).

2.Zhao, Y. Q., S. D. Ding, Y. Yuan, and Y. Z. Wang. "Enhanced Degradation Stability of Poly( P- Dioxanone) under Different Temperature and Humidity with Bis-( 2,6-Diisopropylphenyl) Carbodiimide." Journal of Applied Polymer Science (2014)

Thank you and best regards.

Yours sincerely,

Jizong Zhao

E-mail: zhaojizong@bjtth.org

Reviewer 2 Report

A facile method to design and fabricate 3D-printed bioresorbable stents with PPDO and its composite were introduced in this work, and their fabrication details, morphology, mechanical, biocompatibility properties were also investigated in this manuscript. The overall structure and research strategy are clear and outlined. However, there are some vague data, unnecessary claims, and inappropriate comparisons in the draft. The following revisions are indispensable/strongly recommended to add or revise later for the publication of this manuscript in Polymers

  • Line 27. What’s the reason to lead the current BRSs used for coronary revascularization are not suitable for cerebrovascular revascularization? What are the different requirements for those two applications? What are the current limitations among BRSs, in order to provide your design a chance to overcome? You may want to explain with a brief introduction here, or in section 1
  • Line 61, The reason to “take antiplatelet therapy after the procedure” is not due to the vessel wall injury. If I understand correctly, the implanted, metallic stents are the reasons to lead coagulation and blood clots. Double-check this and make a proper correction.
  • Line 99. Your PPDO stent is designed to degrade & bioresorbed in vivo, correct? Why does it become a remarkable drawback and great limitation?
  • About Figure 1:

1) You should explain your figure in the caption and context, with the details to illustrate each step. The equation or related parameters should be illustrated in Section 2.2, rather than caption here;

2) I don’t understand why the volumetric CT imaging is different from the stent design? If I understand correctly, your stent should be designed and customized with the reference of the lumen, instead of the aneurysm.

3) Can your stent be bent or compressed in the catheter?

  • Line 132, your proportion was derived from literature, please cite this reference
  • Line 139, after washing and drying, what kind of status is the PPDO or PPDO/Stabaxol? Is it a chunk, film, or particles? Since I noticed in Line 145, it became pellets. So have you done any grinding process to create those pellets?
  • Line 149. How do you get the stents off the rod? Is this stent rigid or flexible? How about the adhesion between molten strands and rolling rod? If it's good, is it hard to get them off when they cool down to room temperature? What's the purpose of drying?
  • Line 172, do you have any data or results about your stents’ recovering ability? How long does it take to fully recover to its original dimension? I didn’t see the supporting data in your result
  • Line 178, what’s the mechanical property you used to define the radial compression property? If you use compression modulus, how do you calculate the area, in order to obtain the stress?
  • Line 180-181, what’s the justification to select those two strain values?
  • Line 296, “small pores” on the SEM image, Is it a random defect during the 3D printing? If it’s not, what’s the justification for the formation of such pores?
  • Line 304: How do you calculate the vessel coverage? And what’s the reference value to compare?
  • Section 3.3: How do you tell your stents with high flexibility? Any bending test data? What’s the reference value of compression modulus? How can we tell if it’s better or similar to commercial stents or PLLA stents? Without the comparison (no matter from your own data or reference data), it’s hard to convince that they have “higher” flexibility or “better” radial compression performance
  • Figure 3A. The title for Y-axis should be compression stress if the unit is MPa; And your result didn’t cover the compression modulus, although you mentioned it in section 2.4, correct? For Figure 3C&D, why do you choose 30% of origin diameter? The previous, mechanical data is adapted from 50% of origin. Do you have the stress distribution at that level?
  • Line 331: Does this relative humidity matter? I assume you immersed your sample in the PBS completely and RH should not be impactful.
  • Line 338: have you recorded any mechanical property changes during 8 weeks? From SEM, the pristine PPDO seems more fragile and it will be better with quantitative data to demonstrate the role of Stabaxol
  • Line 341: What can we learn from water absorption? With more water absorption, does it have a faster degradation rate? What’s the reason behind that?
  • Line 392: “better mechanical property”, typo, and how do you define this “better”, do you have any comparison data to support this “better” performance? If you don’t have it on your hand, can you find it from other scholars’ work, like reference[9]?
  • Line 394-395, the compressive properties were enhanced? I didn’t see any significant change based your Figure 3A
  • Line 421, “better flexibility”, again, you need a comparison data to say better flexibility
  • Line 431, typo, radial FORCE
  • Line 432:force ranges from 2 to 4N, 1) which displacement or strain level did those stents record their radial force at? You need to make sure your data is also comparable within the similar setup/conditions;  2) the length of your sample may also impact the force significantly.
  • Line 436: Recovery data: where’s your recovery data? And how long time did it take to recover to the original status?

Author Response

Dear Editor and Reviewers from the Editorial Board:

Thank you for suggesting some revisions and thanks for the reviewers’ comments concerning our manuscript entitled “3D-printed poly (p-dioxanone) stent for endovascular application: in vitro evaluations”. (Manuscript ID: polymers-1651615). We are grateful for your positive comments. We have studied the comments carefully and have made revisions accordingly, which we hope will meet with your approval. The revised sections are marked up using the “Track Changes” function in the paper. The point-by-point responses to the reviewer's comments are as follows:

Point 1: Line 27. What’s the reason to lead the current BRSs used for coronary revascularization are not suitable for cerebrovascular revascularization? What are the different requirements for those two applications? What are the current limitations among BRSs, in order to provide your design a chance to overcome? You may want to explain with a brief introduction here, or in section 1.

Response 1: Thank you for your valuable question and suggestion. we have explained with a brief introduction in section 1(Line 81):“coronary artery disease  usually presents only ischemic lesions, while cerebrovascular disease includes both ischemic and hemorrhagic lesions. The time to complete resorption of most PLLA BRSs for coronary revascularization is more than 2 years, which may be too long for cerebrovascular revascularization with hemorrhagic lesions. Also, cerebrovascular are more tortuous than cardiovascular. The poor compliance of the current PLLA BRSs makes surgical procedures more difficult. Thus, we aim to design a chance to overcome the current limitations among BRSs.”

Point 2: Line 61, The reason to “take antiplatelet therapy after the procedure” is not due to the vessel wall injury. If I understand correctly, the implanted, metallic stents are the reasons to lead coagulation and blood clots. Double-check this and make a proper correction.

Response 2: Thank for your comment. The metallic implants and vessel wall injury both lead to coagulation and blood clots. Thus, patients take antiplatelet therapy after the procedure to avoid the steno-occlusive change of the vessels. We checked this and made a proper correction (Line 62).

Point 3: Line 99. Your PPDO stent is designed to degrade & bioresorbed in vivo, correct? Why does it become a remarkable drawback and great limitation?

Response 3: Thank for your comment. Actually, the degradation duration of PPDO is too fast to match the in-stent endothelialization time. Thus, directly using PPDO as the material of BRS cannot meet the requirement of degradation time in clinical applications. Therefore, the short degradation time of PPDO may be limitation for its use as a BRS material and we have corrected the inaccurate expression in our article. (Line 107).

Point 4: About Figure 1:

1) You should explain your figure in the caption and context, with the details to illustrate each step. The equation or related parameters should be illustrated in Section 2.2, rather than caption here;

2) I don’t understand why the volumetric CT imaging is different from the stent design? If I understand correctly, your stent should be designed and customized with the reference of the lumen, instead of the aneurysm.

3) Can your stent be bent or compressed in the catheter?

Response 4: 1) Thank you for your advice, we explain our figure in the caption and context with the details to illustrate each step. The equation and related parameters have been illustrated in Section 2.2 (Line 172). 2) In this work, our stent is designed and customized with the reference of the lumen and the CT imaging is used to measure the lumen of the vessel. 3) Due to the good flexibility of the catheter material we used, we assumed that the stent can be bent or compressed in the catheter. But we did not carry out any characterization and test to evaluate shape change of stent in the catheter, which will be a research point in the follow-up study.

Point 5: Line 132, your proportion was derived from literature, please cite this reference.

Response 5: Thank you for the suggestion, we cite this reference (Line 155).

Liu, Z. P., S. D. Ding, Y. J. Sui, and Y. Z. Wang. "Enhanced Hydrolytic Stability of Poly(P-Dioxanone) with Polycarbodiimide." Journal of Applied Polymer Science, (2009).

Point 6: Line 139, after washing and drying, what kind of status is the PPDO or PPDO/Stabaxol? Is it a chunk, film, or particles? Since I noticed in Line 145, it became pellets. So have you done any grinding process to create those pellets?

Response 6: Thank you for reviewer’s careful reading. After washing and drying, the status of the PPDO/stabaxol®-1 is a chunk. We then ground the chunk to get PPDO/stabaxol®-1 pellets.

Point 7: Line 149. How do you get the stents off the rod? Is this stent rigid or flexible? How about the adhesion between molten strands and rolling rod? If it's good, is it hard to get them off when they cool down to room temperature? What's the purpose of drying?

Response 7: Thank you for your careful reading. After the printing, we put the rolling rod in the refrigerator to make the molten strands cool and form. We can remove the flexible stent from the rolling rod easily by spraying ethanol. Finally, the 3D-printed PPDO stents were dried at 37℃ for 2 days to remove moisture from the above process (Line 169).

Point 8: Line 172, do you have any data or results about your stents’ recovering ability? How long does it take to fully recover to its original dimension? I didn’t see the supporting data in your result.

Response 8: Special thanks for your suggestion. After 30s unloading the pressure, all BRSs can return to the initial diameter. We have added this results in the Section 3.3 (Line 349).

Point 9: Line 178, what’s the mechanical property you used to define the radial compression property? If you use compression modulus, how do you calculate the area, in order to obtain the stress?

Response 9: Thank you for your careful reading. We used the compression modulus to define the radial compression property of the BRS and calculate the area of the largest cross-section along the long axis of the BRS to obtain the stress.

Point 10: Line 180-181, what’s the justification to select those two strain values?

Response 10: Thanks for your comment. Under the GB/T 1041-2008 (Plastic-Determination of compressive properties), the compression modulus was calculated by the following formula:

Compression modulus (MPa) = Stress difference (σ21)/Strain difference (ε21=0.0025 - 0.0005)

Where σ1 is the corresponding stress value measured when ε1 is set as 0.0005, and σ2 represents the stress value when ε2 is 0.0025.

Point 11: Line 296, “small pores” on the SEM image, Is it a random defect during the 3D printing? If it’s not, what’s the justification for the formation of such pores?

Response 11: Special thanks to you for your good comments. The “small pores” on the SEM image is a random defect during the 3D printing. However, these “small pores” only locate on the surface of the BRS, which maybe not affect the mechanical property of the BRS.

Point 12: Line 304: How do you calculate the vessel coverage? And what’s the reference value to compare?

Response 12: Thanks for this comment. The vessel coverage is the ratio of strut coverage area to stent length vessel area. The reference value to compare is the data of currently commercially available neuro-interventional metal stents.

Point 13: Section 3.3: How do you tell your stents with high flexibility? Any bending test data? What’s the reference value of compression modulus? How can we tell if it’s better or similar to commercial stents or PLLA stents? Without the comparison (no matter from your own data or reference data), it’s hard to convince that they have “higher” flexibility or “better” radial compression performance.

Response 13: Actually, by reason of the small sizes and special shape of 3D printed stents, it is difficult to carry out the three-point bending test in this case. Hence, we did not obtain quantitative data on the flexibility of the BRS and it is also not sufficient to draw the conclusion that the PPDO stent s is better or similar to commercial stents or PLLA stents. We thank the reviewer’s valuable suggestion and have revised the related discussion in article.(Line 346) In addition, to some extent, Figure 2A and the results of the recovery test somewhat are able to demonstrate the flexibility of our BRS.

Point 14: Figure 3A. The title for Y-axis should be compression stress if the unit is MPa; And your result didn’t cover the compression modulus, although you mentioned it in section 2.4, correct? For Figure 3C&D, why do you choose 30% of origin diameter? The previous, mechanical data is adapted from 50% of origin. Do you have the stress distribution at that level?

Response 14: We appreciate for reviewer’s careful reading and suggestions earnestly. The title for Y-aixs in Figure 3A should be compression modulus, we have corrected it (Line 360). For Figure 3C&D, the neo-Hookean strain energy potential W? can be used to estimate the strain energy for this hyperelastic materials only if the deformation does not exceed 30% (Line 262). Therefore, we choose 30% of origin diameter. Also, we wonder whether the theoretical model is still valid when compressed to 50%. Thus, we compress the BRS to 50% of origin in the compression test.

Point 15: Line 331: Does this relative humidity matter? I assume you immersed your sample in the PBS completely and RH should not be impactful.

Response 15: We appreciate for reviewer’s careful reading and suggestions earnestly. We immersed our sample in the PBS completely for degradation experiments. As a result, the relative humidity (RH) refers to the environmental conditions of the laboratory and t has no effect on the degradation behavior of sent. We are sorry to make this ambiguous expression and have corrected it (Line 373).

Point 16: Line 338: have you recorded any mechanical property changes during 8 weeks? From SEM, the pristine PPDO seems more fragile and it will be better with quantitative data to demonstrate the role of Stabaxol.

Response 16: We didn’t record any mechanical property changes during 8 weeks in this work. But it is indeed a worthwhile suggestion to present a quantitative data to demonstrate the role of stabaxol®-1. We appreciate for reviewer’s careful reading earnestly.

Point 17: Line 341: What can we learn from water absorption? With more water absorption, does it have a faster degradation rate? What’s the reason behind that?

Response 17: For most biodegradable materials, especially synthetic polymer materials, hydrolysis is the main mode of degradation; polymer chemical chain type, pH, copolymer composition and water absorption can affect its hydrolysis reaction. The increase of water absorption somewhat demonstrates the beginning of degradation and the increase in the degradation rate. The hydrolysis reaction is a biomolecular reaction between water and active bonds containing functional groups. Hydrophobic polymers absorb less water and degrade slowly and on the contrary, hydrophilic polymers can absorb a lot of water and degrade quickly. Therefore, water absorption is an important index to evaluate the degradation rate.

Point 18: Line 392: “better mechanical property”, typo, and how do you define this “better”, do you have any comparison data to support this “better” performance? If you don’t have it on your hand, can you find it from other scholars’ work, like reference[9]?

Response 18: Thank you for your comment. It has been reported that PLLA has a strength at break close to 50 MPa and the elongation ranging from 3% to 5%, while PPDO has good mechanical properties with a strength at break close to 46 MPa and an ultimate elongation ranging from 500% to 600%. Thus, we stated that the PPDO BRSs has a better mechanical property compared to the PLLA BRS (Line 442).

Relevant references are as follows:

  1. Ren Z, Dong L, Yang Y. Ren, Z. Dong, L. & Yang, Y. Dynamic mechanical and thermal properties of plasticized poly(lactic acid). Journal of Applied Polymer Science, 2006, 101(3):1583-1590.
  2. Baiardo, M., Frisoni, G., Scandola, M., Rimelen, M., Lips, D., Ruffieux, K. and Wintermantel, E. (2003), Thermal and mechanical properties of plasticized poly(L-lactic acid). Journal of Applied Polymer Science, 2003, 90: 1731-1738.

Point 19: Line 394-395, the compressive properties were enhanced? I didn’t see any significant change based your Figure 3A.

Response 19: Thanks to the reviewer’s comments. In Figure 3A the compression force of BRS was slightly increased by adding the stabaxol®-1, though the difference is not significant enough. Our description may be not accurate and we have corrected it (Line 444).

Point 20: Line 421, “better flexibility”, again, you need a comparison data to say better flexibility.

Response 20: Thanks for your advice. As explained in response 13, our description may be not accurate, we have revised the statement “better flexibility” (Line 471).

Point 21: Line 431, typo, radial FORCE.

Response 21: Thank for your comment. We have have corrected it (Line 482).

Point 22: Line 432:force ranges from 2 to 4N, 1) which displacement or strain level did those stents record their radial force at? You need to make sure your data is also comparable within the similar setup/conditions; 2) the length of your sample may also impact the force significantly.

Response 22: We thank the reviewer’s valuable comment and suggestion. Admittedly, the radial force under different measurement conditions and sample lengths is not comparable. However, there is few relevant detailed description of the current commercially available stent mechanics testing methods. As a result, in the study, we performed the test according to the method mentioned in the previous literature (Zhao F, Wang F, Liu L, et al. Composite Self-Expanding Bioresorbable Stents With Reinforced Compression Performance: A Computational and Experimental Investigation[J]. Journal of Vascular Surgery, 2018, 68(3):e82-e83.). Comparing our results with existing data, although not reasonable, it also provides relevant data for reference.

Point 23: Line 436: Recovery data: where’s your recovery data? And how long time did it take to recover to the original status?

Response 23: Thanks for your comments. We have added figure of the recovery of BRS after compression in the supplementary materials (Line 349).

Thank you and best regards.

Yours sincerely

Jizong Zhao

E-mail: zhaojizong@bjtth.org

Reviewer 3 Report

Dear authors,

In general, a very good article,

there are some comments:

paragraph 2.1. line 132, existing literature, reference needs to be added.

paragraph 2.3. line 157, Not the most correct term "crystallization properties", please replace.

paragraph 3.3. It is necessary to add the original S-S curves in the article.

So is PPDO  an amorphous or semi-crystalline polymer (as indicated in the introduction)? The absence of a melting peak must then be explained. Fig. 2.e The curves are very poorly visible, should be enlarged. 

Author Response

Dear Editor and Reviewers from the Editorial Board:

Thank you for suggesting some revisions and thanks for the reviewers’ comments concerning our manuscript entitled “3D-printed poly (p-dioxanone) stent for endovascular application: in vitro evaluations”. (Manuscript ID: polymers-1651615). We are grateful for your positive comments. We have studied the comments carefully and have made revisions accordingly, which we hope will meet with your approval. The revised sections are marked up using the “Track Changes” function in the paper. The point-by-point responses to the reviewer's comments are as follows:

Point 1: Paragraph 2.1. line 132, existing literature, reference needs to be added.

Response 1: Special thanks to reviewer’s careful reading and good advice. We have added the reference in the manuscript (Line 155):

Liu, Z. P., S. D. Ding, Y. J. Sui, and Y. Z. Wang. "Enhanced Hydrolytic Stability of Poly(P-Dioxanone) with Polycarbodiimide." Journal of Applied Polymer Science, (2009).

Point 2: Paragraph 2.3. line 157, Not the most correct term "crystallization properties", please replace.

Response 2: Thank you for your careful reading. We have modified the sentence as the reviewer’s comment and we replace the "crystallization properties" with "crystallinity" (Line 184).

Point 3: Paragraph 3.3. It is necessary to add the original S-S curves in the article

Response 3: Thanks for your comment and advice. We derive the mechanical data from the original curve but we did not save the original S-S curve during the test. It is a negligence. In the study, radial compression tests were conducted using a universal testing machine (KT23.104, Guangzhou Kiatest equipment, Co. Ltd, China) under the international test standard (ISO25539-2-2012). The presser foot first went down to compress stents with the speed of 1 mm/min until 50% deformation of outer diameter. However, since the compression modulus of each BRS was automatically calculated by the testing machine software using the following formula:

Compression modulus (MPa) = Stress difference (σ21)/Strain difference (ε21=0.0025 - 0.0005)

Where σ1 is the corresponding stress value measured when ε1 is set as 0.0005, and σ2 represents the stress value when ε2 is 0.0025.

Point 4: So is PPDO an amorphous or semi-crystalline polymer (as indicated in the introduction)? The absence of a melting peak must then be explained. Fig. 2.e The curves are very poorly visible, should be enlarged.

Response 4: PPDO is an amorphous or semi-crystalline polymer. However, with the increase of the crystallinity, the difficulty of processing the material also increase. In order to fabricate the PPDO BRS, we choose the more amorphous PPDO for printing. Thus, our DCS test failed to detect its melting peak. As the reviewer suggested, we enlarged Figure 2C (Line 325).

Thank you and best regards.

Yours sincerely

Jizong Zhao

E-mail: zhaojizong@bjtth.org

Round 2

Reviewer 1 Report

Thanks for authors's response. ALL researches and  results were clearly presented.